# C2H2 Zinc-Finger Transcription Factors Coordinate Hormone–Stress Crosstalk to Shape Expression Bias of the Flavonoid Pathway in Safflower (*Carthamus tinctorius* L.)

**DOI:** 10.3390/cimb47121023

**Published:** 2025-12-08

**Authors:** Yue Chang, Abdul Wakeel Umar, Minghui Ma, Yuru Zhang, Naveed Ahmad, Xiuming Liu

**Affiliations:** 1College of Life Sciences, Engineering Research Center of the Chinese Ministry of Education for Bioreactor and Pharmaceutical Development, Jilin Agricultural University, Changchun 130118, China; 2Institute for Safflower Industry Research of Shihezi University/Pharmacy College of Shihezi University/Key Laboratory of Xinjiang Phytomedicine Resource and Utilization, Ministry of Education, Shihezi 832003, China; 3Guangdong-Hong Kong Joint Laboratory for Carbon Neutrality, Jiangmen Laboratory of Carbon Science and Technology, Jiangmen 529199, China

**Keywords:** *C2H2* transcription factors, spatiotemporal expression, hormonal and abiotic stresses, flavonoid biosynthesis, *Carthamus tinctorius* L.

## Abstract

C2H2-type zinc-finger transcription factors (ZFPs) play essential roles in plant stress signaling and development; however, their putative functions in safflower have not been systematically characterized. Leveraging the reference genome of the safflower cultivar ‘Jihong-1’ (*Carthamus tinctorius* L.), we investigated the C2H2 family and identified 62 *CtC2H2* genes. Comparative phylogeny with Arabidopsis revealed six subfamilies characterized by shared features such as exon–intron organization and conserved QALGGH motif. Promoter analysis identified multiple light- and hormone-responsive cis-elements (e.g., G-box, Box 4, GT1-motif, ABRE, CGTCA/TGACG), suggesting potential multi-layered regulation. RNA-seq and qRT-PCR analysis identified tissue-specific candidate genes, with *CtC2H2*-*22* emerging as the most petal-specific (6-fold upregulation), alongside *CtC2H2-02*, *CtC2H2*-*23*, and *CtC2H2-24* in seeds (~3-fold), and *CtC2H2*-*21* in roots (3-fold). Under abiotic stresses, *CtC2H2* genes also demonstrated rapid and dynamic responses. Under cold stress, *CtC2H2* genes showed a rapid temporal pattern of expression, with early increase for genes like *CtC2H2*-*45* (>4-fold at 3–6 h) and a delayed increase for *CtC2H2*-*23* at 9 h. A majority of *CtC2H2* genes (8/12) were upregulated by ABA treatment, with *CtC2H2-47* suggesting 3.5-fold induction. ABA treatment also led to a significant increase (2.5-fold) in total leaf flavonoid content at 24h, which is associated with the significant upregulation of flavonoid pathway genes *CtANS* (5-fold) and *CtCHS* (3.3-fold). Simultaneously, UV-B radiation induced two distinct expression patterns: a significant suppression of four genes (*CtC2H2*-*23* decreased to 30% of control) and a complex fluctuating pattern, with *CtC2H2*-*02* upregulated at 48 h (2.8-fold). MeJA elicitation revealed four complex expression profiles, from transient induction (CtC2H2-02, 2.5-fold at 3 h) to multi-phasic oscillations, demonstrating the functional diversity of CtC2H2-ZFPs in jasmonate signaling. Together, these results suggest stress and hormone-responsive expression modules of C2H2 ZFPs for future functional studies aimed at improving stress adaptation and modulating specialized metabolism in safflower.

## 1. Introduction

Safflower (*Carthamus tinctorius* L.) is a widely cultivated member of the Asteraceae family valued in traditional Chinese medicine and as an oilseed crop, is reported to exhibit hypolipidemic [1], hypoglycemic [2], anti-inflammatory [3], antitumor [4,5], antithrombotic and neuroprotective activities [6]. Major metabolites include flavonoids, phenolic acids, and quinones. Among these, Flavonoids are multifunctional secondary metabolites, which have important functions in resistance to various stresses [5,7], mainly as antioxidants and regulators of plant polar growth hormone transport (PAT) [8]. Flavonoids are able to accumulate at the level of epidermal cell layers as protective UV radiation absorbers during UV-B stress [9,10], as well as acting as food repellents in the face of herbivores [11]. Flavonoids tend to increase antioxidant activity, promote photosynthesis, influence stomatal activity and regulate root growth, thereby enhancing plant resistance to abiotic stress [5,12,13]. When plants are exposed to cold stress, they are able to accumulate flavonoids in large quantities to increase their cold resistance and reduce cold damage [14].

Zinc-finger proteins (ZFPs) constitute a large transcription factor family in plants and are associated with growth, hormone signaling, and responses to environmental stress. ZFPs were first identified in *Xenopus laevis* oocytes [15] and contain a canonical Cys2/His2 motif (X_2C-X_2–4C-X_12H-X_2–8H) [16]. The tetrahedral “zinc finger” is formed when two Cys and two His residues chelate a zinc ion, stabilizing a β-hairpin/α-helix structure. ZFPs are classified by the number and spacing of these residues, including C2H2 (TFIIIA), C3HC4 (RING), C4 (GATA-type), and others [17]. The C2H2 subgroup constitutes one of the largest transcription-factor families in plants and is intimately linked to growth regulation. These genes are closely associated with plant development, hormonal signaling, and tolerance to biotic and abiotic stresses. Their proteins typically harbor a QALGGH DNA-binding motif, an EAR repression motif, a nuclear-localization B-box, a DLN transcriptional-regulation box, and a leucine-rich L-box [18]. Prior studies indicate that C2H2 transcription factors contribute to stress tolerance (e.g., salinity and cold responses) and developmental processes, including trichome formation and flowering-time regulation [19,20,21]. Genome-wide studies now span numerous crops, including Arabidopsis [22], alfalfa [23], tomato [24], apple [25], rice [26], pepper [27], cucumis [28], potato [29], ginseng [30], lotus [31], and sweet potato [32]. Recent functional studies further show that overexpressing C2H2-ZFPs improves drought resilience in poplar [33], enhances root growth under osmotic stress in wheat [34], and modulates apple drought responses [35], highlighting their importance in biotechnological applications. However, the specific members of the CtC2H2 family involved, their response to hormonal and environmental cues, and their direct regulatory influence on the flavonoid pathway in safflower remain largely unexplored.

Despite extensive work in model and crop species, the *C2H2* gene family has not been systematically characterized in safflower. Here, we provide the first genome-wide identification and systematic characterization of the *CtC2H2* TF family. We identify 62 *CtC2H2* genes in safflower and conduct a comprehensive analysis of their sequence features, phylogeny, and expression patterns under abiotic stresses and hormonal treatments, with a particular focus on flavonoid-related ABA responses. The primary objectives of this study were to classify the *CtC2H2* family through phylogeny and motif analysis, to investigate their spatio-temporal and stress-responsive expression patterns, and to unravel the potential coordination between *CtC2H2* TFs, ABA signaling, and the flavonoid biosynthesis pathway. By establishing this foundational framework, our work pinpoints key candidate *CtC2H2* genes for future molecular breeding aimed at developing stress-resilient safflower cultivars with enhanced flavonoid content.

## 2. Materials and Methods

### 2.1. Plant Material and Abiotic-Stress Treatments

The Jihong No.1 safflower cultivar was germinated in moist trays and grown under controlled conditions (25 °C, 40% relative humidity, 16 h light/8 h dark photoperiod) in a growth chamber. After 3 days of germination, the plants were cultured for 4 weeks, and uniformly sized seedlings were selected for several stress assays, including ultraviolet-B irradiation (UV-B irradiation (W/m^−2^), abscisic acid (ABA, 200 μM), methyl jasmonate (MeJA, 200 μM), and low temperature exposure (4 °C) [36]. MeJA and ABA were both prepared as stock solutions in anhydrous ethanol and diluted to 0.1% ethanol solutions for use. During spraying, plant leaves were thoroughly wet but not dripping. The control group received an equal volume of 0.1% ethanol solution and was cultivated under the same growth conditions. Seedlings were sprayed with the respective treatment solutions or sterile water (control) and maintained under the same growth conditions. For UV-B, samples were harvested at 0, 6, 12, 24, 36 and 48 h; for the other stresses at 0, 3, 6, 9, 12 and 24 h. True leaves from four-week-old safflowers were selected as samples. For each time point, three biological replicates were rapidly frozen in liquid nitrogen and stored at −80 °C for subsequent RNA extraction and metabolite analysis.

### 2.2. Identification of C2H2 Genes in Carthamus Tinctorius

Protein sequences of Arabidopsis thaliana C2H2-ZFPs were downloaded from TAIR (https://www.arabidopsis.org/ (accessed on 18 July 2024)) and used as queries in a bidirectional BLASTP search against the “Jihong-1” safflower genome [37], implemented in TBtools II (Version 2.0). To obtain a comprehensive candidate list, we first screened plant C2H2-type zinc-finger proteins reported in the literature and then retrieved the Hidden Markov Model (HMM) profile for the C2H2 domain (PF00096) from Pfam 33.1 (https://pfam.xfam.org/ (accessed on 1 August 2024)). The PF00096 HMM was used in HMMER3 (v3.3) [38] to search the safflower proteome, complementing the BLASTP approach and ensuring recovery of highly divergent members. Putative proteins were analyzed using the NCBI Conserved Domain Search to confirm the presence of the canonical Cys_2_-His_2_ domain prior to classification as CtC2H2 family members. Physicochemical parameters, including molecular weight (MW), theoretical isoelectric point (pI) and grand average of hydropathicity (GRAVY) were calculated with ProtParam (http://web.expasy.org/protparam/ (accessed on 29 August 2024)). Sub-cellular localization was predicted in WoLF PSORT (https://wolfpsort.hgc.jp/ (accessed on 30 July 2024)). Finally, redundant isoforms were removed, and domain-validated proteins were cataloged as CtC2H2 family members for downstream analyses.

### 2.3. Conserved Motifs and Gene Structure Organization of the CtC2H2-ZFPs

Conserved motifs were identified with MEME Suite v5.5.3 (https://meme-suite.org/ (accessed on 15 September 2024)) [39]) using a maximum of 15 motifs (width = 6–60 aa). The MEME suite (Version 5.5.8) was used under the following key parameters: the motif distribution was set to “Zoops” and low-complexity filtering was enabled. Sequence logos were rendered in WebLogo 3 [40]. Coding sequences were aligned to their genomic loci, and exon–intron arrangements were extracted in TBtools v2.315 [41]. Motif and gene-structure diagrams were integrated side-by-side in TBtools for comparative visualization.

### 2.4. Phylogenetic Analysis of CtC2H2s Family

In order to perform comparative phylogenetic analysis, Protein sequences of Arabidopsis thaliana C2H2-ZFPs were included as reference taxa. Multiple-sequence alignment and neighbor-joining phylogeny were generated in MEGA 11 [42]. The phylogenetic tree was constructed using the Jones-Taylor-Thornton (JTT) model, with pairwise deletion applied to gaps and missing data. The reliability of the tree was assessed with 1000 bootstrap replications. The phylogenetic tree was edited and color-coded in EvolView v2 (https://www.evolgenius.info/evolview-v2/ (accessed on 1 October 2024)) [43] to further improve the quality of the tree.

### 2.5. Cis-Regulatory Elements Analysis in CtC2H2 Promoters

Based on the nucleotide sequences 2000 bp upstream of the transcription start site (TSS) of the 62 CtC2H2 genes downloaded from the safflower Genome Database. Promoter regions (2000 bp upstream of the transcription start site) were analyzed using PlantCARE (Plant Cis-Acting Regulatory Element) [44] (http://bioinformatics.psb.ugent.be/webtools/plantcare/html/ (accessed on 15 October 2024)) to identify cis-acting regulatory elements. The identified regulatory elements were categorized based on their functional roles, including light responsiveness, hormone regulation, and stress-related motifs [45]. The results and organization of cis-acting elements were visualized using TBtools [41].

### 2.6. Functional Annotation Analysis of CtC2H2 Genes

Gene Ontology (GO) annotations were assigned using Blast2GO using an E-value significance threshold of 1.0E-3 [46], categorizing *CtC2H2* genes into biological process, cellular component, and molecular function terms. The Gene Ontology (GO) enrichment analysis was conducted to categorize the genes based on their functional roles, including but not limited to transcriptional regulation, zinc ion binding, and response to abiotic stress [47]. The enrichment was calculated using Fisher’s Exact Test, with a False Discovery Rate (FDR) significance threshold of <0.05. The data were processed, and the corresponding graphs, including bar plots and scatter plots representing GO term distributions, were plotted using R 4.3.0 [48] with the ggplot2 package 3.4.0. [49].

### 2.7. RNA Extraction and Real-Time Quantitative PCR

During the expression validation phase, we screened these genes and selected 12 candidate genes that were rich in elements responsive to cold, light, methyl jasmonate and abscisic acid. Gene-specific primers were designed using Primer3Plus [50] and synthesized by Bioengineering (Changchun) Co., Ltd. (Changchun, China) (primers listed in Appendix A). Total RNA was extracted from samples using RNAiso Plus reagent (Takara Bio, Shiga, Japan) following the manufacturer’s protocol. RNA quality and concentration were verified using a NanoDrop spectrophotometer (Thermo Fisher Scientific, Wilmington, DE, USA; Software version 3.8.0) and agarose gel electrophoresis. First-strand cDNA was synthesized from 1 μg of total RNA using ToloScript All-in-one RT EasyMix for qPCR (TOLOBIO, Shanghai, China), with oligo (dT) primers in a 20 μL reaction volume.

Quantitative real-time PCR (qRT-PCR) was performed on a QuantStudio 3 Real-Time PCR System (Applied Biosystems, Thermo Fisher Scientific, Waltham, MA, USA) using 2 × Q5 SYBR qPCR Master Mix (Universal) (TOLOBIO Biotech, Shanghai, China). Each reaction mixture (20 μL total volume) contained: 2 × Q5 SYBR qPCR Master Mix (10 μL); Gene-specific forward and reverse primers (0.4 μL each, 10 μM); cDNA template (1 μL); Nuclease-free dH_2_O (8.2 μL). The qRT-PCR analysis was performed using standard cycling conditions (95 °C for 30 s, followed by 40 cycles of 95 °C for 10 s and 60 °C for 30 s). Melting-curve analysis was used to verify amplicon specificity [51]. All reactions were performed in technical triplicates, along with no-template controls (NTCs) for each primer pair. The expression data were analyzed using the 2^−ΔΔCt^ method [52], with Ct60S (KJ634810) employed as the internal reference gene for normalization [53]. Relative gene expression levels were calculated and presented as fold-changes compared to control samples.

### 2.8. Tissue-Specific CtC2H2 Gene Expression and Validation

To examine the spatial and temporal expression patterns of the *CtC2H2* gene family, we analyzed the transcriptome data of safflower across multiple developmental stages [54]. The expression profiles (FPKM values) of *CtC2H2* genes were examined in various tissues including root, stem, leaf, bud, primordial, full bloom, decline, 10-day seedlings, 20-day seedlings, and 30-day seedlings. Hierarchical clustering analysis was performed, and heat maps were plotted to visualize expression patterns using TBtools II (Version 2.0) software [41], with a red to blue color gradient indicating relative transcript abundance high to low levels, respectively. This visualization approach effectively revealed tissue-specific and developmentally regulated expression patterns among *CtC2H2* family members [55]. For experimental validation, twelve *CtC2H2* genes containing cis-acting regulatory elements associated with hormone-responsive pathways were selected for quantitative real-time PCR analysis according to the previously discussed method [51].

### 2.9. Determination of Total Flavonoid Content and Detection of Key Flavonoid-Biosynthetic Genes

A 0.10 g aliquot of freeze-dried leaf tissue was finely ground and extracted with 2 mL of HPLC-grade methanol. The suspension was sonicated (40 kHz) at 50 °C for 1 h, cooled on ice, and centrifuged at 12,000× *g* for 10 min. The supernatant was filtered through a 0.22 µm PTFE syringe filter (Merck Millipore, Billerica, MA, USA) and stored at −20 °C until analysis. Chromatographic separation and quantification were performed on an Agilent 1200 HPLC (Agilent Technologies, Santa Clara, CA, USA) equipped with a ZORBAX 300SB-C18 column (Agilent Technologies, Santa Clara, CA, USA) (5 µm, 4 m × 5 m). The isocratic mobile phase consisted of methanol: 0.4% (*v*/*v*) formic-acid water = 1:1 at 1 mL min^−1^. The detection was set at 360 nm and 25 °C. Rutin (Solarbio, Beijing, China) served as the external standard which was eluted at a retention time of 5.22 min with a symmetric peak shape and clear baseline separation from potential impurities. A five-point calibration curve (0.01–0.20 mg mL^−1^; R^2^ > 0.999) was used to express results as mg rutin g^−1^ DW [56]. Three biological replicates and three technical injections were analyzed for each treatment. Expression of key flavonoid-biosynthetic genes (*CtCHS*, *CtCHI*, *CtF3H*, *CtF3′H*, *CtFLS*, *CtDFR*, *CtANS*) was quantified by qRT-PCR exactly as described in 2.8; primer sequences are provided in Appendix A. Relative transcript levels were normalized to *Ct60S* and calculated with the 2^−ΔΔCt^ method [52].

### 2.10. Statistical Analysis

The qRT-PCR fold changes were calculated using the 2^−ΔΔCt^ method [52]. For each experimental treatment, three independent biological replicates were analyzed, with each replicate measured in three technical repetitions. The data from the technical replicates were averaged to represent a single biological replicate. All numerical values are expressed as the mean ± standard deviation of these three biological replicates. Data processing and graph generation were performed using GraphPad Prism v8.0 (GraphPad Software, San Diego, CA, USA). Statistical significance was determined by one-way ANOVA followed by Tukey’s HSD post hoc test, with significance levels denoted as * *p* < 0.05, ** *p* < 0.01, and *** *p* < 0.001. For datasets that violated the assumption of homogeneity of variances, we have replaced the analysis with the more robust Welch’s ANOVA followed by the Games–Howell post hoc test.

## 3. Results

### 3.1. Identification of CtC2H2 Genes and Analysis of Physicochemical Properties

A total of 62 C2H2-type zinc-finger transcription factors were retrieved from the safflower genome by BLASTP screening and subsequent verification with the NCBI’s Batch CD-Search software (accessed on 1 August 2024; with the default parameters (E-value threshold 0.01, CDD database version 3.20) to confirm the presence of the canonical Cys_2_/His_2_ domain. These genes were designated *CtC2H2-01-62* (Appendix A). Protein lengths span 134–1191 aa, with calculated molecular weights of 15.57–132.76 kDa and theoretical pI values of 4.84–10.91. Instability indices vary from 31.1 to 85.9; only *CtC2H2-17*, *-19*, *-53*, and *-57* fall below the empirical stability threshold (II < 40). Aliphatic indices range between 47.64 and 79.93, whereas all CtC2H2 proteins exhibited negative grand average hydropathicity (GRAVY) scores ranging from −1.137 to −0.305, indicating hydrophilic properties. The WoLF PSORT webtool classified every CtC2H2 protein as nuclear, which may reflect their function as transcription factors [57]. The CtC2H2 complement (*n* = 62) is close to the 58-member C2H2 set recently documented in apple [25] yet significantly smaller than the 99 genes identified in tomato [24], highlighting lineage-specific expansion among dicotyledons.

### 3.2. Phylogenetic Analysis

To clarify the evolutionary relationship between safflower and Arabidopsis C2H2-type zinc-finger proteins, we constructed a phylogenetic tree from 62 CtC2H2 proteins and 39 Arabidopsis C2H2 members using MEGA 11 (neighbor-joining algorithm, 1000 bootstrap replications) [42]. A total of six phylogenetic subfamilies (Ⅰ–Ⅵ) were identified based on conserved domain architecture and sequence similarity (Figure 1). These classifications were mainly consistent with the clades distribution reported for tomato [24] and apple [25]. Subfamily counts were as follows: I, 12 genes; II, 12 genes; III, 5 genes; IV, 16 genes; V, 10 genes; and VI, 7 genes, together accounting for all 62 CtC2H2 members. This pattern suggests lineage-specific gene expansion, particularly in subfamily IV, which harbors > 25% of the total family and may reflect adaptive diversification in safflower.

### 3.3. Conserved Motifs and Gene Structural Features of the CtC2H2-ZFPs

In order to reveal the diversity of the CtC2H2 family, 15 conserved motifs in its full-length protein sequence were analyzed according to the phylogenetic tree using MEME and TBtools, and the 15 motifs analyzed were named motif 1–15 (Figure 2). Among them, motif 1 is present in 62 CtC2H2 proteins, suggesting that it is highly conserved and potentially important in the CtC2H2 family. In addition, other motifs have inter-family similarities. For example, subfamilies I, II, and III all contain motif 1 and motif 2, and subfamilies IV and V all contain motif 1 and motif 7. motif 1 and motif 2 both contain the QALGGH sequence, a core sequence, and motif 5 contains the DLNL sequence, an EAR motif that has been shown to play an important role in transcriptional repression and abiotic stress response (Appendix A). Similarly, motif 3 contains EXEXXAXCLXXL (L-box), a leucine-rich region thought to play an important role in protein–protein interactions. Among the C2H2-ZFPs identified, four of the 12 members of subfamily I are intron-deficient; there are no introns in subfamilies II and V; among the five members of subfamily III, one intron is missing; among the 16 members of subfamily IV, four are intron-deficient; and among the seven members of subfamily VI, five are intron-deficient. Among them, subfamily IV has the highest proportion of intron-deficient genes (71.4%).

**Figure 2 cimb-47-01023-f002:**
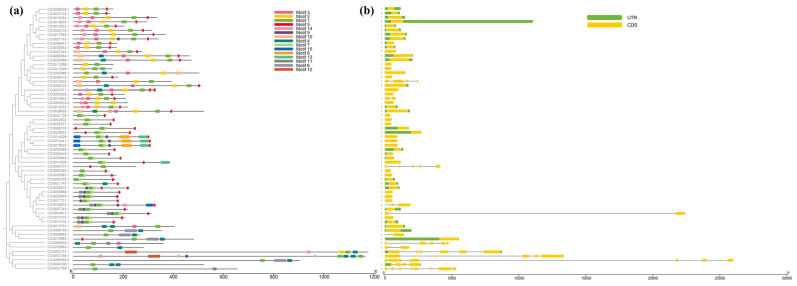
Structural features of the *CtC2H2* zinc finger gene family in safflower. (**a**) Conserved motif composition of CtC2H2 proteins identified using MEME suite. Colored boxes represent distinct conserved motifs, and their relative positions and arrangements indicate potential functional divergence among CtC2H2 members. (**b**) Gene structure organization of CtC2H2 family members, showing the distribution of exons, introns, and untranslated regions (UTRs). Exons are represented by colored boxes, introns by black lines, and UTRs by gray boxes. The comparison of motif distribution and gene structure highlights evolutionary conservation and structural diversity within the CtC2H2 family.

### 3.4. Analysis of Cis-Acting Elements of CtC2H2 Genes Promoters

The promoter regions of 62 *CtC2H2* genes upstream of the 2000 bp fragment in safflower were analyzed, and the results showed that the cis-acting elements of the safflower *C2H2* promoter region could be classified into three major categories: growth and developmental elements, hormone-responsive elements, and biotic and abiotic stress elements (Figure 3). Most of the safflower *C2H2* was enriched with light-responsive elements, such as G-box, Box 4, and GT1-motif, suggesting that safflower *C2H2* may be involved in the regulation of light signaling pathways that affect plant photomorphogenesis. Most *CtC2H2* promoters contain hormone responsive elements, such as ABRE (abscisic acid responsive element), CGTCA-motif (methyl jasmonate responsive element), TGACG-motif (methyl jasmonate responsive element). In addition, the promoter sequences of *CtC2H2* genes contained elements related to biotic and abiotic stress response such as drought, flooding, and cold responsive elements. Several *CtC2H2* members also contained multiple MYB binding sites, suggesting regulatory potential of safflower *C2H2* genes related to growth and development and may play important roles in stress response.

**Figure 3 cimb-47-01023-f003:**
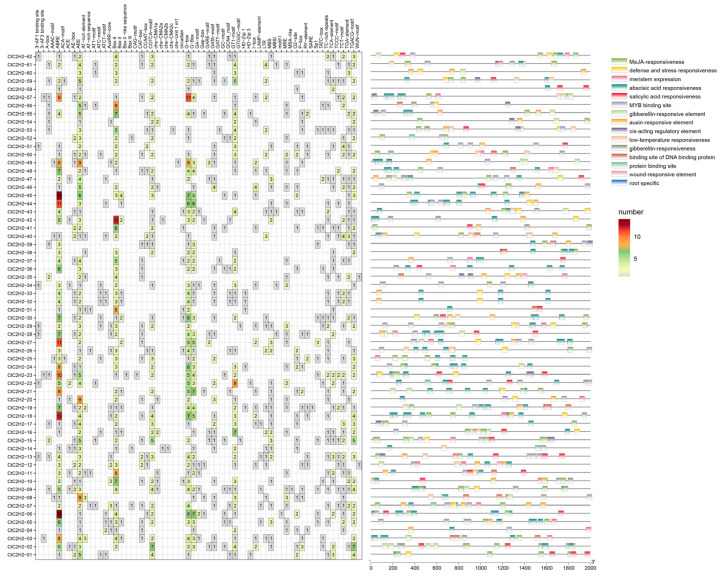
Analysis of cis-acting elements of the safflower *C2H2*-type zinc finger protein family promoter. The cis-acting elements were predicted in the promoter sequences of the *CtC2H2* genes. Rectangular boxes of distinct colored boxes represent the different types of cis-acting elements.

### 3.5. Functional Categorization and Annotation Enrichment of CtC2H2 Gene Family

To further understand the functional relationships among the *CtC2H2s* gene family members, the *CtC2H2* gene family was analyzed by GO functional annotation for functional annotation (Figure 4). Among the 62 transcripts, 62 genes were annotated to at least one of the three major functions: 27 were annotated to biological process (BP), 38 to cellular component (CC), and 61 to molecular function (MF). Eighteen of these transcripts were annotated for only one function, 26 transcripts were annotated with two major functions, and 18 transcripts were annotated with three major functions. At Level 2 (Figure 4b), seven sub-levels were enriched in BP: regulation of biological process (GO:0050789), negative regulation of biological process (GO:0048519), response to stimulus (GO:0050896), cellular process (GO:0009987), multicellular organismal process (GO:0032501), biological regulation (GO:006500), developmental process (GO:0032502); one sub-level was enriched as a cellular anatomical structure (GO:0110165) in CC; and two sub-levels were enriched in MF sub-levels for binding (GO:0005488), transcription regulator activity (GO:0140110). GO functional classification and enrichment suggested diverse transcriptional regulation and stress responsiveness, indicating potential regulatory functions.

### 3.6. Tissue-Specific Expression Analysis of C2H2 Family Genes in Safflower

In order to study the expression pattern of the C2H2 zinc finger protein family genes in safflower, the transcriptome data of safflower cultivar “Jihong No.1” were analyzed at the root, stem, leaf, flower and seed stages. As shown in Figure 5a, the transcriptome data revealed distinct tissue-specific expression profiles. Notably, *CtC2H2*-*02*, *CtC2H2*-*23*, and *CtC2H2*-*24* were highly expressed in seeds, with *CtC2H2*-*02* showing approximately 3-fold increased expression compared to other tissues. Similarly, *CtC2H2*-*06* and *CtC2H2*-*22* were highly specific to petals during the fading stage, exhibiting peak expressions of 3-4-fold and 6-fold greater than in leaves, respectively. In contrast, *CtC2H2*-*10* was markedly upregulated in leaves (2-fold), while *CtC2H2*-*21*, *CtC2H2*-*29*, and *CtC2H2*-*47* showed strong root-specific expression, with *CtC2H2*-*21* reaching 3-fold enrichment in roots. Finally, *CtC2H2*-*35* demonstrated its highest expression in stems, at 2-fold higher than in other tissues. Based on the presence of stress and hormone-responsive cis-acting elements, twelve *CtC2H2* genes were selected for experimental validation via qRT-PCR (Figure 5b). The trends of these 12 genes in stems, leaves, and flowers were consistent with the transcriptome data, proving the accuracy of the transcriptome data.

### 3.7. Cold Stress Triggers Differential and Dynamic Regulation of CtC2H2-ZFPs

The gene expression analysis was carried out in the leaves of safflower seedlings treated with cold stress (4 °C) at different time points (3, 6, 9, 12, 24 h), and normalized expression was demonstrated using normothermic treatment (25 °C) as the 0 h control. As shown in Figure 6, the five genes *CtC2H2-02*, -*06*, -*15*, -*22*, and -*45* exhibited a transient expression pattern, peaking at different time points. These five genes reached the peak expression after 3, 6 or 12 h of cold treatment, and decreased thereafter. Similarly, a subset of seven *CtC2H2* genes exhibited a diverse pattern, reaching maximum expression at 6–9 h before undergoing a secondary upregulation phase at 12–24 h. However, the expression trend of *CtC2H2-29* gene exhibited transient induction followed by partial downregulation over time, although its overall expression was lower than that of the control group. The dissociation curve analysis for qRT- PCR primer specificity validation are given in (Appendix A). Together, these findings suggested that *CtC2H2* genes responded differently to cold treatments, indicating diverse regulatory roles in cold-stress responses.

### 3.8. CtC2H2-ZFPs Exhibit Distinct Induced and Repressed Expression Profiles Under UV-B Exposure

We further investigated the expression profile of *CtC2H2* genes in the leaves of safflower seedlings exposed to UV-B stress at multiple time points 0, 3, 6, 9, 12, and 24 h. The results showed that *CtC2H2* genes showed distinct induced and repressed expression patterns under the exposure of UV-B stress at variable time periods. For example, the expression level of four genes such as *CtC2H2-10*, *CtC2H2-22*, *CtC2H2-23* and *CtC2H2-24* were significantly suppressed at a consistent pattern when treated with UV-B stress at almost every time point. In contrast, a more complex and dynamic expression pattern was observed for the remaining *CtC2H2* genes. For instance, the expression pattern of *CtC2H2-15*, *CtC2H2-21*, *CtC2H2-29*, *CtC2H2-35*, and *CtC2H2-45*, showed upregulation at 6 h when compared to the control group, suggesting the immediate response of these genes to UV-B perception (Figure 7). However, the expression of *CtC2H2-02* showed maximum increase at a further 48 h timepoint than the control. Together, the diverse induction of CtC2H2 genes suggested their specialized role in the short longer-term adaptive or repair processes, which is potentially involved in the restoration of cellular homeostasis after UV-B stress.

### 3.9. Differential Expression Responses of CtC2H2-ZFP Genes Under ABA Treatments

To investigate the role of *CtC2H2* genes under hormonal treatment, we investigated their expression following treatment with 200 μM ABA at different time periods (0, 3, 6, 9, 12, and 24 h). As shown in Figure 8, ABA treatment significantly induced the expression of several *CtC2H2* genes, suggesting their potential involvement in ABA-responsive pathways. For instance, a group of eight *CtC2H2* genes (*CtC2H2-02*, -*15*, *-21*, *-22*, *-23*, *-24*, *-35*, *-45*) displayed a consistent pattern: their expression first increased at 3 or 6 h, then decreased, and finally increased again at 9 or 12 h after ABA treatment. On the other hand, the expression level of three genes (*CtC2H2-06*, *-10*, *-47*) decreased to a minimum at 6–9 h timepoint and then recovered (Figure 8). However, the expression peaks of *CtC2H2-06* and *-10* never rose above control levels, whereas *CtC2H2-47* showed a significant peak higher than the control. Together, these results suggested both induced and repressed *CtC2H2* members and define their specific temporal expression profiles, providing a foundation for understanding their specialized functions in ABA signaling.

### 3.10. Distinct and Complex Expression Profiles of CtC2H2-ZFPs Under MeJA Stress

To further extend the investigation of *CtC2H2* genes expression in the leaves of safflower seedlings under hormonal treatment, we carried out their expression profiling MeJA treatment at different time periods (0, 3, 6, 9, 12, and 24 h). The results showed that genes like *CtC2H2-02* and *CtC2H2-47* demonstrated a rapid initial upregulation at 3–6 h, and a subsequent decline to at 9 h (Figure 9). In addition, *CtC2H2-06* and *CtC2H2*-*29* exhibited a single, transient wave of expression suggesting increased and/or stable expression at 3 h, followed by a significant and continuous decrease from 6 h onwards. On the contrary, most of *CtC2H2* genes including *CtC2H2-10*, *-15*, *-21*, *-22*, *-23*, *-24*, and *-35*, showed periodic enhanced expression at 3 h, 6 h, and 12 h (Figure 9). While *CtC2H2-45* exhibited a unique pattern of steady decrease after MeJA treatment, only reaching its peak expression at 24 h. These results suggest the MeJA treatment induced expression of a subset of *CtC2H2* genes in a time-dependent manner.

### 3.11. Changes in Total Flavonoids Content and Key Genes of the Flavonoid Pathway Under ABA Treatment

As shown in Figure 7 above, the expression of most safflower *CtC2H2-ZFPs* increased significantly at 24 h timepoint under 200 μmol/L ABA treatment. Therefore, we tend to extract the total flavonoids content from untreated (0 h) versus ABA-treated 24 h safflower leaves. As demonstrated in Figure 10, following a 24 h exogenous ABA treatment, a substantial and highly significant accumulation (paired *t*-test, *p* < 0.001) of total flavonoid content was observed in safflower, with a notable increase from 0.56 mg/g DW to 1.39 mg/g DW, representing an approximate 2.5-fold enhancement. In line with this observation, the expression levels of multiple pivotal genes within the flavonoid biosynthesis pathway underwent substantial alterations. Among them, the expression of *CtANS* and *CtCHS* was most significantly increased, with 5.0-fold and 3.3-fold increases, respectively (both *p* < 0.001). Furthermore, *CtFLS* and *CtDFR* expression levels increased significantly by 1.8-fold (*p* = 0.001) and 1.5-fold (*p* < 0.01), respectively. In contrast, the expression of *CtF3H* and *CtF3′H* was found to be significantly repressed, with levels decreasing to 14% and 41% of the control levels, respectively (*p* < 0.001 for both). These findings suggest that ABA effectively promotes flavonoid synthesis and accumulation in safflower by synergistically enhancing the expression of *CtCHS*, *CtFLS*, *CtDFR*, and *CtANS* genes, while concurrently suppressing the expression of *CtF3H* and *CtF3′H* genes. These correlations suggest that *CtC2H2* genes may influence flavonoid biosynthesis, potentially through ABA signaling pathways.

## 4. Discussion

C2H2-type zinc-finger proteins (ZFPs) represent one of the important families of transcriptional factors in plants. This largest but poorly explored transcription factors are known to mediate responses to developmental cues and environmental stresses in plants [33,34,35]. Studies have shown that members of *C2H2* demonstrated reduced sensitivity to ABA in apple, and improves developmental cues in Arabidopsis under drought stress [35]. Similarly, C2H2-type ZFPs in tobacco and soybean showed enhanced cold tolerance responses by regulating the ABA-induced expression of cold-responsive genes [58]. Prior studies also suggested the potential role of C2H2-type ZFPs in regulating ABA and MeJA mediated abiotic stress response in a variety of plant species [59,60,61]. The current study presents the genome-wide identification *CtC2H2* in safflower and elucidates their distinct expression bias under abiotic stress (cold and UV-B) and hormonal treatment (ABA and MeJA).

### 4.1. C2H2-Type ZFPs Expression Bias Under Cold Stress

The investigation of our expression analysis revealed distinct temporal patterns among CtC2H2 genes under cold stress, with individual genes exhibiting peak expression at different time intervals (Figure 6). This dynamic expression suggests a tightly regulated transcriptional network activated in response to cold stress. Such observations are consistent with earlier studies emphasizing the pivotal role of *C2H2* genes in orchestrating cold-responsive expression [62,63,64]. The transient expression profiles observed in our study further support the notion that multiple transcription factors function in a coordinated manner to regulate plant cold tolerance. As reported in several prior studies, C2H2-type ZFPs are widely recognized as central regulators of cold stress responses, primarily through ABA-mediated signaling cascades [65,66]. Plant species including Arabidopsis [67], soybean [36], and rice [65] demonstrated that the overexpression of key C2H2 ZFPs led to cold-responsive (COR) genes via ABA-responsive promoter elements, promoted osmolyte biosynthesis, and improved ion homeostasis. For instance, SCOF-1 and members of the AZF/STZ family enhance cold tolerance by strengthening ABA-regulated transcriptional pathways, while ZFPs such as GmZF1 and ZFP182 elevate COR gene expression, proline accumulation, and other protective metabolites that mitigate cellular damage [67]. In line with these findings, *CtC2H2* genes in safflower highlight the conserved role of this transcription factor family in cold acclimation across plant species (Figure 6). It is well established that ABA accumulates rapidly upon exposure to cold, subsequently activating downstream genes enriched in ABA-responsive cis-elements [68]. The regulation imparted by *CtC2H2* genes likely contributes to this adaptive cascade by fine-tuning stress-responsive transcription, thereby facilitating essential physiological adjustments such as osmotic balance, membrane stabilization, and oxidative stress mitigation. Collectively, our results suggest that safflower employs a complex regulatory framework involving temporally dynamic *CtC2H2* gene activation to enhance cold acclimation.

### 4.2. Regulation of C2H2-Type ZFPs Expression Under UV-B Stress

The differential expression patterns of *CtC2H2* genes under UV-B exposure also led us towards understanding the complexity of transcriptional regulation activated during UV-induced stress (Figure 7). Our findings revealed that while several *CtC2H2* genes were consistently repressed across all time points, a substantial subset of genes showed dynamic and time-dependent induction (Figure 7). This divergence suggests that C2H2-type ZFPs function in distinct regulatory modules that collectively contribute to UV-B perception, signaling, and downstream acclimation responses. For instance, the continuous suppression of *CtC2H2*-*10*, *CtC2H2*-*22*, *CtC2H2*-*23*, and *CtC2H2*-*24* suggested that these genes may act as negative modulators of UV-B stress-responsive pathways or participate in growth-associated processes that are downregulated under UV-B stress to conserve energy and minimize damage. On the contrary, genes such as *CtC2H2*-*15*, *CtC2H2*-*21*, *CtC2H2*-*29*, *CtC2H2*-*35*, and *CtC2H2*-*45* exhibited rapid induction at the 6 h time point, indicating an early activation phase potentially linked to UV-B signal transduction, ROS scavenging, or initiation of protective mechanisms. The strong late induction of *CtC2H2*-*02* at 48 h further reflects a distinct role in long-term adaptive or repair responses, possibly associated with DNA damage recovery, membrane stabilization, or restoration of cellular homeostasis.

Comparable temporal expression biases of *CtC2H2* and other classes of transcription factors families have been reported in other plant species, which mainly contribute to UV-B tolerance by modulating antioxidant pathways, DNA repair mechanisms, and flavonoid biosynthesis [69,70,71]. The varied responses observed in safflower suggested that *CtC2H2* genes participate at multiple stages of the UV-B stress response, from early perception to late acclimation, indicating their functional diversity within this regulatory landscape (Figure 7). Collectively, the induction–repression module suggests that safflower employs a finely tuned transcriptional network of C2H2-ZFPs to mitigate UV-B-induced damage and maintain physiological stability under elevated radiation.

### 4.3. Hormonal-Induced Regulatory Modules of CtC2H2-ZFPs

The diverse expression patterns of *CtC2H2* genes following ABA and MeJA treatments highlight the multifaceted regulatory roles of C2H2-type ZFPs in hormone-mediated stress responses. We also observed that ABA treatments induced notable transcriptional adjustments and coordination with flavonoid biosynthesis across multiple *CtC2H2* members, reflecting their potential roles in ABA-dependent signaling pathways. For example, under ABA treatment, a distinct cluster of genes including *CtC2H2-02*, *-15*, *-21*, *-22*, *-23*, *-24*, *-35*, and *-45*, demonstrated early induction at 3–6 h, a decline at mid-points, and a subsequent secondary increase at 9–12 h. This dual-peak profile suggests a possible involvement in both early ABA perception and later downstream regulatory events, potentially mediating transcriptional reprogramming associated with stress adaptation, stomatal regulation, or osmotic adjustment. Conversely, *CtC2H2-06*, *-10*, and *-47* were initially downregulated, reaching minimum expression at 6–9 h; however, only *CtC2H2-47* significantly regained its expression level above control levels. Such contrasting responses imply functional specialization, where certain C2H2-ZFPs may act as positive regulators of ABA-responsive pathways while others serve as negative modulators, fine-tuning ABA signal amplitude to prevent excessive or prolonged responses. This distinct regulatory pattern of expression aligns with findings in Arabidopsis [67], rice [65], and soybean [67], where C2H2-type ZFPs are known to modulate ABA-triggered processes such as osmolyte biosynthesis, ROS detoxification, and transcriptional activation of stress-responsive genes.

In comparison, MeJA treatment also triggered more heterogeneous and transient expression patterns across the *CtC2H2* gene family. Early peaks observed in *CtC2H2-02* and CtC2H2-47, along with the single-wave responses of *CtC2H2-06* and *-29*, suggest rapid JA-mediated activation followed by attenuation, possibly reflecting their involvement in defense-related signaling or early JA perception. Additionally, most genes including *CtC2H2-10*, *-15*, *-21*, *-22*, *-23*, *-24*, and *-35*, exhibited periodic induction at multiple time points, suggesting repeated engagement in JA-regulated transcriptional cycles. The distinctive late-stage peak of CtC2H2-45 further suggests roles in prolonged or secondary MeJA responses, potentially linked to longer-term defense or stress recovery mechanisms. Consistent with our findings, previous studies have shown that the expression of ZPT2-3 is activated through JA-dependent signaling pathways but ethylene-independent mechanisms, highlighting the sensitivity of C2H2-ZFPs to JA-mediated regulatory cues [72,73]. Moreover, JA can function together with ethylene through integrators such as ERF1, further contributing to the modulation of stress-responsive transcriptional networks [74]. These observations support the diverse and time-dependent induction patterns of *CtC2H2* genes under MeJA treatment in our study, indicating that select CtC2H2-ZFPs may act as important components of JA-driven defense and stress adaptation pathways in safflower.

Although MeJA influenced a broad subset of *CtC2H2* genes, the ABA-induced transcriptional shifts were more prominent, structured, and consistent across family members. This emphasizes the centrality of ABA signaling in orchestrating C2H2-ZFP-mediated responses in safflower. The presence of early-response spikes, mid-phase suppression, and late-response surges under ABA treatment strongly suggests that *CtC2H2* genes integrate temporal ABA signals to mediate finely tuned regulatory programs. Such coordinated transcriptional dynamics may reflect the involvement of C2H2-ZFPs in ABA-regulated processes such as drought and cold tolerance, stomatal behavior, metabolic adjustment, and ABA-responsive gene activation via ABRE-containing promoters. Collectively, the combined hormone-responsive expression landscape underscores the functional diversity of *CtC2H2* genes, with ABA emerging as the dominant hormonal regulator. These insights highlight the potential of specific *CtC2H2* members as key molecular nodes for enhancing ABA-mediated stress resilience and provide a foundation for future functional characterization and breeding strategies aimed at improving hormone-responsive stress tolerance in safflower.

### 4.4. ABA-Induced Flavonoid Accumulation and Potential Regulatory Roles of CtC2H2-ZFPs

In this study, the observed increase in total flavonoid content following ABA treatment provides compelling evidence that ABA-induced regulation of *CtC2H2* functions as a key positive regulator of flavonoid biosynthesis in safflower. The ~2.5-fold increase in total flavonoids after 24 h of exogenous ABA application indicates a strong metabolic reprogramming toward enhanced secondary metabolite production. This response aligns with well-established findings in other plant species, where ABA acts as a central signaling molecule that modulates stress-responsive metabolic pathways, including those associated with antioxidant and protective compounds such as flavonoids [75,76]. The transcriptional upregulation of major biosynthetic genes including *CtCHS*, *CtFLS*, *CtDFR*, and *CtANS*, further reinforces the notion that ABA promotes flux through the flavonoid pathway induced by *CtC2H2* expression as observed in other plant species [77,78]. The marked increases in *CtANS* and *CtCHS* expression suggest an activation of early and late biosynthetic steps, ultimately contributing to higher flavonoid accumulation. Importantly, the strong induction of *CtC2H2*-ZFPs observed at 24 h under ABA treatment suggests that these transcription factors may act upstream or in parallel with flavonoid biosynthesis genes [79]. C2H2-type ZFPs are known to participate in ABA-mediated transcriptional cascades, often influencing genes involved in metabolic adjustment and stress adaptation [80,81]. The temporal correlation between *CtC2H2* upregulation and enhanced flavonoid biosynthesis implies that certain members of this family may modulate the expression of key enzymes in the flavonoid pathway, either directly through promoter binding or indirectly via ABA-responsive regulatory circuits. Such interactions would be consistent with the broader roles of C2H2-ZFPs in stress signaling, ROS management, and secondary metabolism reported in other plant species. Collectively, these findings highlight a coordinated ABA-driven mechanism in safflower, wherein CtC2H2-ZFPs and flavonoid biosynthetic genes jointly contribute to elevated flavonoid accumulation. Such dynamic responses emphasize the need for further investigation into these regulatory networks, especially considering their implications for metabolic engineering aimed at enhancing stress resilience in crops.

## 5. Conclusions

This study presents the first comprehensive genome-wide identification of C2H2 zinc-finger transcription factor family in safflower. The structural conservation and expression bias of candidate *CtC2H2* genes uncovered distinct and tissue-specific expression modules under basal, abiotic stresses (cold, UV-B) and hormonal treatments (ABA, MeJA). The coordinated upregulation of specific *CtC2H2* genes with key flavonoid pathway genes under ABA treatment suggests their potential role in regulating specialized metabolism. Collectively, these findings highlight the functional diversity of CtC2H2-ZFPs and provide valuable candidate genes for future functional studies aimed at enhancing stress resilience and modulating the production of valuable metabolites in safflower.

## Figures and Tables

**Figure 1 cimb-47-01023-f001:**
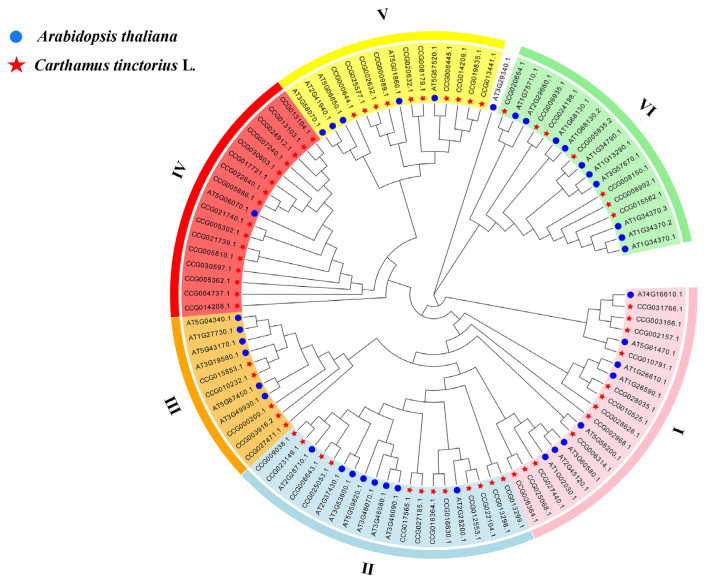
Phylogenetic tree of C2H2 proteins in Arabidopsis thaliana and *Carthamus tinctorius*. The C2H2 protein sequences of the two species were aligned by MEGA X and the tree was built with the NJ method. The tree was further categorized into six distinct subfamilies in different colors. Proteins were grouped into six subfamilies (Ⅰ–Ⅵ) based on conserved domain composition and sequence similarity. Each subfamily is color-coded: I (pink), II (light blue), III (orang), IV (red), V (yellow), and VI (green).

**Figure 4 cimb-47-01023-f004:**
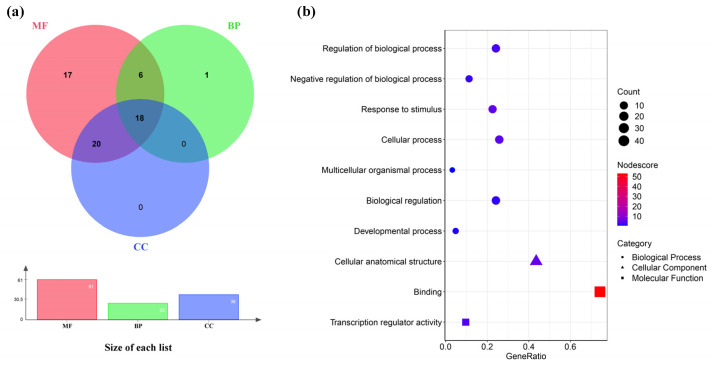
Functional categorization and annotation enrichment analysis of the *CtC2H2* genes. (**a**) A Venn network of the *CtC2H2* genes were among the biological process (BP), cellular component (CC), and molecular function (MF) functional categories; (**b**) The *CtC2H2* gene was categorized into ten Gene Ontology (GO) functional categories: seven biological process (BP) categories (circles), one cellular component (CC) category (triangles), and two molecular function (MF) categories (squares).

**Figure 5 cimb-47-01023-f005:**
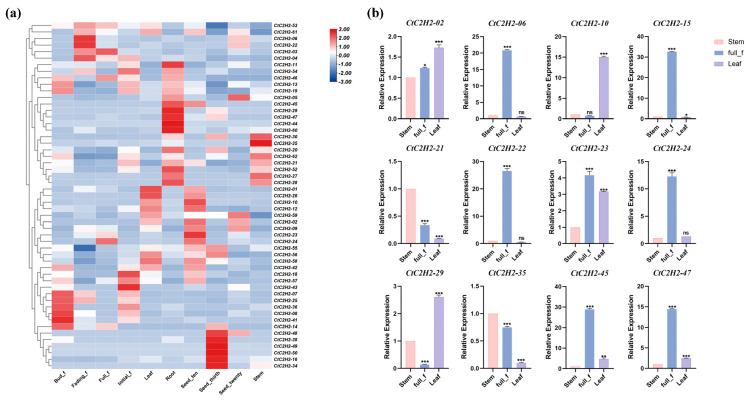
Gene expression analysis of the *CtC2H2-ZFPs*. (**a**) Expression profiles of the safflower *CtC2H2* gene family in different tissues; (**b**) qRT-PCR Validation of Expression Patterns of 12 *CtC2H2* Genes in Roots, Stems, and Petals at Bloom Stage (* *p* < 0.05, ** *p* < 0.01, *** *p* < 0.001; ns = non significant).

**Figure 6 cimb-47-01023-f006:**
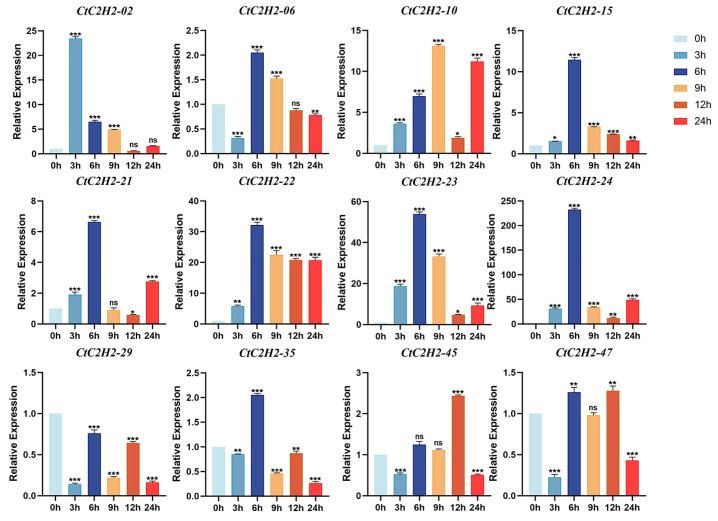
Expression patterns of *CtC2H2-ZFP* genes in safflower under cold stress. Relative transcript levels of selected *CtC2H2-ZFP* genes were analyzed at 0, 3, 6, 9, 12, and 24 h following treatment with 4 °C. Expression values were quantified by qRT-PCR and normalized against the internal control gene. Error bars represent the mean ± standard deviation (SD) of three biological replicates. Asterisks indicate statistically significant differences compared with the 0 h control (* *p* < 0.05, ** *p* < 0.01, *** *p* < 0.001; ns = non significant), revealing distinct temporal expression patterns of *CtC2H2-ZFPs* in response to cold stress.

**Figure 7 cimb-47-01023-f007:**
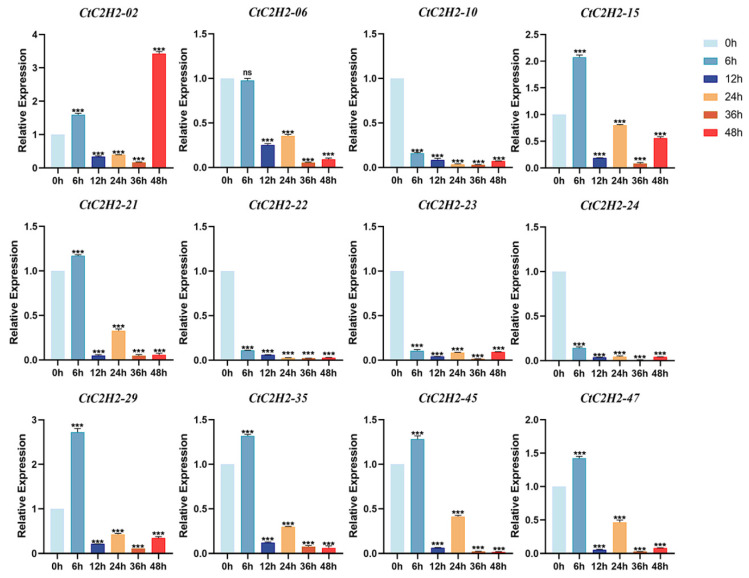
Expression patterns of *CtC2H2-ZFP* genes in safflower under UV-B stress. Relative expression levels of selected *CtC2H2-ZFP* genes were measured at 0, 6, 12, 24, 36, and 48 h following UV-B radiation treatment. Transcript abundance was quantified by qRT-PCR and normalized against the internal reference gene. Error bars represent the mean ± standard deviation (SD) of three biological replicates. Asterisks denote statistically significant differences compared with the 0 h control (*** *p* < 0.001; ns = non significant), illustrating time-dependent transcriptional responses of *CtC2H2-ZFPs* to UV-B-induced stress.

**Figure 8 cimb-47-01023-f008:**
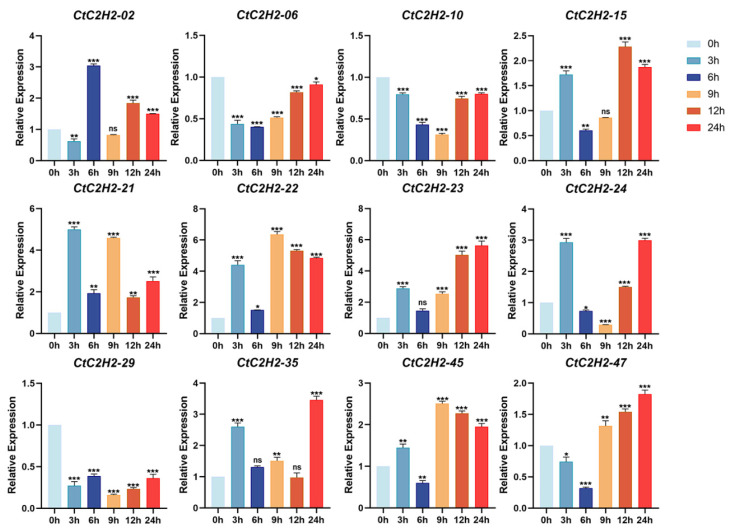
Induced and repressed expression pattern of *CtC2H2-ZFP* genes in safflower under ABA treatment. Relative transcript levels of selected *CtC2H2-ZFP* genes were analyzed at 0, 3, 6, 9, 12, and 24 h following treatment with 200 μmol·L^−1^ abscisic acid (ABA)**.** Expression values were quantified by qRT-PCR and normalized against the internal control gene. Error bars represent the mean ± standard deviation (SD) of three biological replicates. Asterisks indicate statistically significant differences compared with the 0 h control (* *p* < 0.05, ** *p* < 0.01, *** *p* < 0.001; ns = non significant), revealing distinct temporal expression patterns of *CtC2H2-ZFPs* in response to ABA-induced stress.

**Figure 9 cimb-47-01023-f009:**
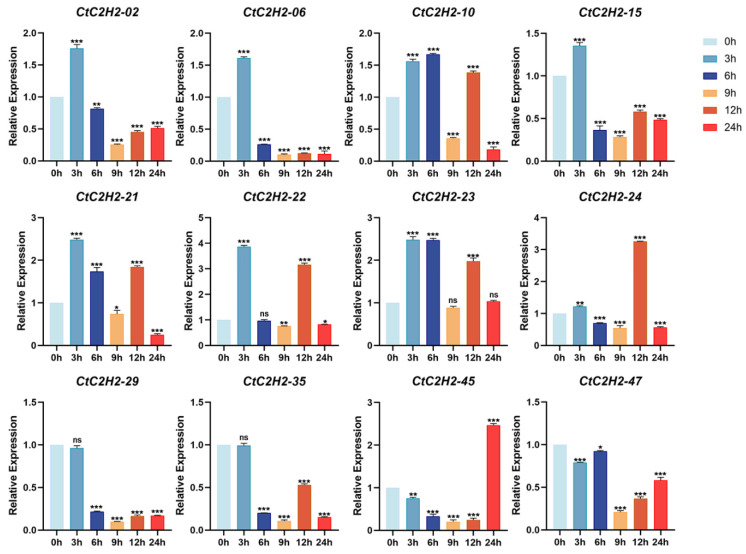
Expression dynamics of *CtC2H2-ZFP* genes in safflower under MeJA treatment. Relative transcript levels of selected *CtC2H2-ZFP* genes were determined at 0, 3, 6, 9, 12, and 24 h following treatment with 200 μmol·L^−1^ methyl jasmonate (MeJA). Gene expression was quantified by qRT-PCR and normalized against the internal control gene. Error bars represent the mean ± standard deviation (SD) of three biological replicates. Asterisks indicate statistically significant differences compared with the 0 h control (* *p* < 0.05, ** *p* < 0.01, *** *p* < 0.001; ns = non significant), highlighting the transcriptional responses of *CtC2H2-ZFPs* to MeJA-induced signaling.

**Figure 10 cimb-47-01023-f010:**
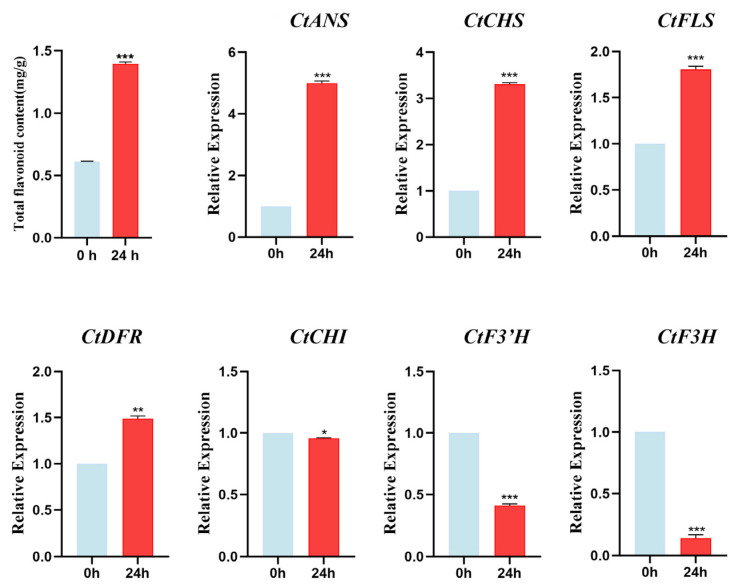
Total flavonoid content and expression profiles of key flavonoid biosynthetic genes under ABA treatment. The upper panel shows the total flavonoid content in safflower was determined at 0 h and 24 h following treatment with 200 μmol·L^−1^ abscisic acid (ABA)**.** Data represent the mean ± standard deviation (SD) of three biological replicates. The lower panel shows relative expression levels of key enzyme genes involved in the flavonoid biosynthesis pathway were analyzed by qRT-PCR at 0 h and 24 h under the same ABA treatment conditions. Transcript levels were normalized against the internal reference gene. Statistical significance determined by paired Student’s *t*-test compared with the 0 h control is indicated (* *p* < 0.05, ** *p* < 0.01, *** *p* < 0.001), revealing ABA-induced modulation of flavonoid accumulation and associated biosynthetic gene expression.

## Data Availability

All data supporting the findings of this study are included in the article and its Appendix A (e.g., Appendix A for gene lists and primers). Materials and any additional datasets (such as raw qRT-PCR Ct values and representative HPLC chromatograms) are available from the corresponding author on reasonable request. The safflower genome sequence was deposited at NCBI under the bioproject accession number: PRJNA399628 and the transcriptome data are publicly available at NCBI under accession number: PRJNA909037.

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
