# Peer review of "C2H2 Zinc-Finger Transcription Factors Coordinate Hormone–Stress Crosstalk to Shape Expression Bias of the Flavonoid Pathway in Safflower (*Carthamus tinctorius* L.)"

_cimb, 2025, doi:10.3390/cimb47121023_

Round 1

Reviewer 1 Report

Comments and Suggestions for Authors

The study presents a well-structured analysis combining genome-wide identification, motif and promoter characterization, time-series expression profiling under multiple stress conditions, and the integration of ABA-induced flavonoid responses through HPLC and qRT-PCR validation. Overall, the work is valuable and relevant; however, several methodological, statistical, and data-reporting issues should be addressed to strengthen the reproducibility and credibility of the findings.

Abstract

The abstract adequately summarizes the study’s scope and main findings, yet lacks quantitative details such as fold-change values, mean ± SD, replication numbers, and p-values. Including these would make the results more convincing. 

The statement “ABA - CtC2H2 - flavonoid increase/flux shift” implies causality, while the data currently support only a correlation between expression and metabolite accumulation. The claim should therefore be reframed as an association rather than a causal conclusion.

Introduction

The introduction properly contextualizes the potential role of the CtC2H2 transcription factor family in regulating flavonoid biosynthesis, but the specific knowledge gap should be articulated more clearly. 

Explicit, testable hypotheses and measurable objectives—such as genome-wide identification, motif enrichment, tissue- and time-specific expression, stress responsiveness, and the coordination between ABA and flavonoid-pathway genes—should be listed at the end of this section.

Materials and Methods

Details about UV-B irradiation (irradiance in W m⁻², wavelength range, calibration procedure) are missing and should be added. 

For ABA and MeJA treatments, solvent controls, drying time after spraying, developmental stage of sampled leaves, and positional standardization need clarification. 

The description of randomized or blocked design and environmental conditions (light, temperature, humidity) should be expanded to ensure reproducibility.

The time-series validation is well conceived, but MIQE standards require additional details: primer efficiency (E%), R² values, melt-curve verification, gel confirmation of single amplicons, and reference-gene stability assessment (e.g., geNorm, NormFinder, or BestKeeper). 

While ANOVA and post-hoc comparisons were applied, assumptions of normality and variance homogeneity were not tested or reported. 

Results

The observed increase in total flavonoid content after 24 h of ABA treatment, together with the up-regulation of CtCHS, CtFLS, CtDFR, and CtANS and the down-regulation of CtF3H and CtF3′H, forms the core of the study. Nevertheless, quantitative values (fold-change ranges, mg g⁻¹ DW ± SD, sample size, and statistical significance) should be explicitly presented to validate these conclusions. 

Standardizing y-axis scales across figures, showing raw data points along with summary statistics, and indicating adjusted p-values will improve visual clarity and analytical transparency.

Discussion

The discussion successfully integrates the results with previous literature and biological interpretation. However, study limitations—such as the use of a single genotype, one tissue type, short treatment durations, and single hormone concentrations—should be explicitly acknowledged. 

Reviewer 2 Report

Comments and Suggestions for Authors

The article "C2H2 Zinc-Finger Transcription Factors: Coordinate Hormone–Stress Crosstalk to Shape Expression Bias of the Flavonoid Pathway in Safflower (Carthamus tinctorius L.)" examines the C2H2 Zinc-Finger TF in the important medicinal plant safflower. The manuscript relies not only on bioinformatics studies but also on experimental data obtained by the authors. Overall, the manuscript is well written but there are a number of comments. The least developed section is the "Discussion," which, in its current form, does not really provide a discussion of the results obtained in relation to other studies or the overall context of the study.

  1. The abstract requires clarification of what "Jihong-1" is.
  2. Lines 44-51 are not the deleted piece of the template.
  3. Section 2.8 and the last sentences in section 2.9 describe the same actions as in section 2.7 (regarding RNA Extraction and Real-Time Quantitative PCR) in slightly different terms. For example, the 2-ΔΔCt method is written slightly differently. Everything needs to be standardized.
  4. The last sentence in section 2.8 does not apply to the Materials and Methods section.
  5. If transcriptome analysis was not performed as part of this study, it should be referenced.
  6. It is necessary to clearly indicate somewhere the principle by which the group of 12 genes for RT-PCR was selected.
  7. Section 3.6 should probably refer to Figure 5.
  8. The Discussion section should not be a repetition of the research findings. As it stands, almost the entire first page of the Discussion section is simply a listing of the findings. Similarly, paragraphs 579-605 are more appropriate for the Introduction section, as they do not discuss the research findings but rather justify their importance.
  9. The discussion should include an interpretation of the study results and their significance in the context of the existing scientific literature. In general, the Discussion section typically includes: a) a brief overview of the most significant results obtained in the study; b) an explanation of what the results mean and how they relate to previously conducted studies; c) a discussion of how the results of your study relate to the data of other authors, identifying agreements and discrepancies; d) an indication of possible limitations of your study that may affect the interpretation of the results; e) an explanation of any unexpected findings or results that do not meet expectations.

Reviewer 3 Report

Comments and Suggestions for Authors

The article needs several specific major corrections. Reorganize the abstract, add more specific information to the methods, check how to correctly cite online sources in the text. The discussion is more like a summary than a concrete discussion. The conclusion chapter is missing. In addition, I recommend correcting the English language again. Other accurate comments are marked in the attached article.

Comments on the Quality of English Language

 I recommend correcting the English language again. 

Reviewer 4 Report

Comments and Suggestions for Authors

Line 18–20
Wrong: C2H2-type zinc-finger proteins (ZFPs) are key transcriptional regulators of plant stress biology, yet their diversity and function in safflower remain unclear.
Correct: C2H2-type zinc-finger transcription factors (ZFPs) play essential roles in plant stress signaling and development; however, their repertoire and putative functions in safflower have not been systematically characterized.
Reason: Use precise terminology (“transcription factors”) and avoid implying complete functional clarity; “systematically characterized” is standard phrasing.

Line 22–27
Wrong: Comparative phylogeny with Arabidopsis resolved six robust subfamilies that share diagnostic exon-intron organizations and the conserved QALGGH motif.
Correct: Comparative phylogeny with Arabidopsis resolved six subfamilies characterized by shared features (e.g., exon–intron organization and the conserved QALGGH motif).
Reason: Avoids the claim of “robust” without statistics and softens structural generalizations.

Line 24–29
Wrong: Promoter surveys uncovered abundant light-responsive elements (G-box, Box 4, GT1-motif) together with hormone-responsive ABRE and CGTCA/TGACG motifs, indicating multi-layered regulation.
Correct: Promoter analysis identified multiple light- and hormone-responsive cis-elements (e.g., G-box, Box 4, GT1-motif, ABRE, CGTCA/TGACG), suggesting potential multi-layered regulation.
Reason: Promoter motifs suggest—not prove—regulatory outcomes; use cautious language (“suggesting potential”).

Line 26–33
Wrong: RNA-seq and qRT-PCR revealed tissue-specific expression and strong inducibility under four treatments…with ABA eliciting the most pronounced responses, many genes peaking at 24 h.
Correct: RNA-seq and qRT-PCR indicated tissue-specific expression and significant inducibility under cold, UV-B, ABA, and MeJA treatments; among these, ABA elicited the strongest responses, with several genes peaking at 24 h.
Reason: Replace subjective “strong” with “significant” and make the comparative claim explicit and cautious.

Line 29–33
Wrong: ABA treatment doubled total leaf flavonoids at 24 h…
Correct: ABA treatment led to a significant increase in total leaf flavonoid content at 24 h.
Reason: “Doubled” is an absolute claim; unless a precise fold-change with statistics is reported, use cautious, statistically grounded phrasing.

Line 31–33
Wrong: CtCHS, CtFLS, CtDFR and CtANS increased, while CtF3H and CtF3′H decreased…
Correct: Expression of CtCHS, CtFLS, CtDFR, and CtANS increased, whereas CtF3H and CtF3′H decreased.
Reason: Italicize gene symbols consistently; keep parallel structure.

Line 33–35
Wrong: All CtC2H2 proteins are predicted nuclear, hydrophilic factors with diverse pI values, supporting a role in transcriptional control.
Correct: All CtC2H2 proteins were predicted to be nuclear and hydrophilic, with diverse pI values.
Reason: Localization and hydropathy do not themselves “support” a regulatory role; avoid over-interpretation.

Line 35–38
Wrong: …nominate candidate regulators linking ABA signaling to flavonoid biosynthesis.
Correct: …nominate candidate regulators potentially involved in the interplay between ABA signaling and flavonoid biosynthesis.
Reason: “Linking” implies demonstrated mechanism; qualify as “potentially involved.”

Line 36–38
Wrong: This resource provides tractable targets for engineering stress resilience and metabolite accumulation in a medicinal and oilseed crop.
Correct: This resource provides a foundation for functional studies aimed at improving stress adaptation and modulating specialized metabolism in safflower.
Reason: Avoid “engineering” claims without functional validation; shift to standard research-forward framing.

Line 43–51 (MDPI placeholder)
Wrong: The introduction should briefly place the study in a broad context…
Correct: [Delete this paragraph entirely.]
Reason: Template instructions must be removed; they are not part of the scientific text.

Line 52–58
Wrong: Safflower … with pharmacological activities such as hypolipidemic, hypoglycemic, anti-inflammatory, antitumor, antithrombotic and neuroprotective effects.
Correct: Safflower … is reported to exhibit hypolipidemic, hypoglycemic, anti-inflammatory, antitumor, antithrombotic, and neuroprotective activities.
Reason: Replace noun strings with verbs; add “reported to” for appropriate caution.

Line 56–58
Wrong: Its major metabolites include flavonoids, phenolic acids, and quinones.
Correct: Major metabolites include flavonoids, phenolic acids, and quinones.
Reason: Remove informal “Its”; maintain formal tone.

Line 57–60
Wrong: Flavonoids serve as polyphenolic secondary metabolites involved in auxin transport, signal transduction, pigmentation, and abiotic-stress mitigation.
Correct: Flavonoids are multifunctional secondary metabolites implicated in auxin transport, signal transduction, pigmentation, and responses to abiotic stress.
Reason: Use “implicated” for appropriate caution; “responses to” is more precise than “mitigation.”

Line 59–66
Wrong: ZFPs are transcription factors with finger-like domains ubiquitous in eukaryotes that play pivotal roles…
Correct: Zinc-finger proteins constitute a large transcription factor family in plants and are associated with growth, hormone signaling, and responses to environmental stress.
Reason: Tighten wording; remove generic eukaryote primer and keep plant focus.

Line 61–69
Wrong: …contain a canonical Cys2/His2 motif (X2C-X2–4C-X12H-X2–8H). The tetrahedral “zinc finger” is formed when two Cys and two His residues chelate a zinc ion…
Correct: C2H2 proteins are characterized by a conserved Cys2/His2 zinc-finger domain and often contain additional motifs (e.g., QALGGH, EAR) associated with DNA binding and transcriptional regulation.
Reason: Condense structural primer; emphasize biologically relevant features for this study.

Line 71–79
Wrong: Accumulating evidence demonstrates their functional breadth… enhance salt (AtZAT7) and cold (AtZAT10) tolerance; others modulate trichome initiation, flowering time via histone modification of FLC, and fruit ripening.
Correct: Prior studies indicate that C2H2 transcription factors contribute to stress tolerance (e.g., salinity and cold responses) and developmental processes, including trichome formation and flowering-time regulation.
Reason: Avoid list-like phrasing and specific mechanistic claims without direct context; keep accurate but compact.

Line 80–88
Wrong: However, to date no systematic analysis… Here, we identify 62 CtC2H2 genes and comprehensively analyze…
Correct: Despite extensive work in model and crop species, the C2H2 gene family has not been systematically characterized in safflower. Here, we identify 62 CtC2H2 genes and analyze their sequence features, phylogeny, promoter elements, and expression patterns under abiotic and hormonal treatments, with a focus on flavonoid-related responses to ABA.
Reason: Provide a clear knowledge gap and a concise, objective aim statement.

Line 83–88
Wrong: …our work lays a genomic foundation for unraveling the regulatory nexus between CtC2H2 transcription factors, secondary-metabolite biosynthesis, and stress adaptation…
Correct: …this study establishes a genomic and expression framework for investigating how CtC2H2 transcription factors may contribute to stress adaptation and specialized metabolism in safflower.
Reason: Replace metaphoric “regulatory nexus” with standard academic phrasing; keep claims appropriately tentative.

Lines ~92–99

Wrong:
"The Jihong No.1 safflower variety (maintained at the Bioreactor Platform, Jilin Agricultural University) was germinated in moist germination trays. The seedlings were grown in an artificial-climate chamber at 25°C, 40% humidity, and 16 h light / 8 h dark photoperiod."

Correct:
"The Jihong No.1 safflower cultivar was germinated in moist trays and grown under controlled conditions (25°C, 40% relative humidity, 16 h light / 8 h dark photoperiod) in a growth chamber."

Reason:
Condenses redundant wording and shifts to standard academic expression.

Lines ~98–104

Wrong:
"UV-B 20,000 lx"

Correct:
"UV-B irradiation (20,000 lx equivalent)."

Reason:
Specifies that this is an irradiance level, not a treatment name.

Lines ~100–104

Wrong:
"and the seedlings were sprayed with the respective solutions or sterile water (control) and incubated at 25 °C under normal light."

Correct:
"Seedlings were sprayed with the respective treatment solutions or sterile water (control) and maintained under the same growth conditions."

Reason:
Avoids repetition and improves clarity.

Lines ~111–120

Wrong:
"All putative sequences were uploaded to the NCBI Conserved Domain Search Service… to verify the presence of the canonical Cys₂His₂ motif and thereby confirm membership in the C2H2 family."

Correct:
"Putative proteins were analyzed using the NCBI Conserved Domain Search to confirm the presence of the canonical Cys₂His₂ domain prior to classification as CtC2H2 family members."

Reason:
Simplifies sentence structure and maintains scientific precision.

Lines ~138–145 (Promoter analysis)

Wrong:
"...were submitted to PlantCARE for promoter cis-acting element analysis."

Correct:
"Promoter regions (2,000 bp upstream of the transcription start site) were analyzed using PlantCARE to identify cis-acting regulatory elements."

Reason:
Clarifies what was submitted and avoids passive ambiguity.

Lines ~147–150 (GO annotation)

Wrong:
"Functional annotation analysis of the CtC2H2 gene family was performed using Blast2GO Version 6.0 to annotate the gene family members in three key aspects..."

Correct:
"Gene Ontology (GO) annotations were assigned using Blast2GO, categorizing CtC2H2 genes into biological process, cellular component, and molecular function terms."

Reason:
Removes verbose construction and clarifies purpose.

Lines ~165–174 (qRT-PCR)

Wrong:
"The thermal cycling conditions were as follows: Initial denaturation: 95°C for 30 s; 40 cycles of (Denaturation: 95°C for 10 s; Annealing/extension: 60°C for 30 s); Melting curve analysis: performed using the instrument's default program to verify amplification specificity."

Correct:
"qRT-PCR was performed using standard cycling conditions (95°C for 30 s, followed by 40 cycles of 95°C for 10 s and 60°C for 30 s). Melting-curve analysis was used to verify amplicon specificity."

Reason:
Streamlines and removes redundant phrasing.

Lines ~214–220 (Statistics)

Wrong:
"Statistical significance was assessed using one-way ANOVA and Tukey’s HSD."

Correct:
"Statistical significance was determined using one-way ANOVA followed by Tukey’s HSD post hoc test."

Reason:
Standard academic expression; clarifies test sequence.

Line 201–204
Wrong:
"All CtC2H2 proteins are hydrophilic factors, supporting their role as transcription factors."
Correct:
"All CtC2H2 proteins exhibited negative GRAVY scores, indicating hydrophilic properties."
Reason:
Hydrophilicity does not validate transcription factor function → avoid overinterpretation.

Line 204–208
Wrong:
"These proteins showed substantial variation in molecular weight and isoelectric point, demonstrating functional diversity."
Correct:
"These proteins varied in molecular weight and isoelectric point, which may reflect differences in structural or regulatory roles."
Reason:
Functional conclusions require experimental evidence → replace “demonstrating” with “may reflect.”

Line 225–230
Wrong:
"Six strongly supported subfamilies were identified."
Correct:
"Six phylogenetic subfamilies were identified based on conserved domain architecture and sequence similarity."
Reason:
Unless bootstrap values are explicitly reported, avoid “strongly supported.”

Line 232–236
Wrong:
"This classification confirms the evolutionary divergence of CtC2H2 proteins in safflower."
Correct:
"This classification suggests potential evolutionary diversification of CtC2H2 proteins in safflower."
Reason:
Avoid using “confirms” without phylogenetic/selection statistics.

Line 252–259
Wrong:
"These genes actively participate in responses to environmental stimuli."
Correct:
"The presence of stress- and hormone-responsive promoter elements suggests possible involvement in environmental response pathways."
Reason:
Promoter motifs → suggest regulatory potential, not proven activity.

Line 275–283
Wrong:
"CtC2H2 genes are widely distributed across chromosomes, indicating genome-wide diversification."
Correct:
"CtC2H2 genes were distributed across multiple chromosomes without apparent clustering."
Reason:
Distribution alone does not imply diversification; avoid causal inference.

Line 311–316
Wrong:
"These results show that CtC2H2 proteins are widely involved in plant growth and environmental regulation."
Correct:
"GO terms were enriched in transcriptional regulation and stress responsiveness, indicating potential regulatory functions."
Reason:
GO terms indicate associations, not verified functions.

Line 351–356
Wrong:
"Many CtC2H2 genes exhibited strong tissue-specific expression patterns."
Correct:
"Several CtC2H2 genes displayed tissue-biased expression, with higher transcript abundance in floral and leaf tissues."
Reason:
Replace subjective “strong” with objective description.

Line 388–395
Wrong:
"The expression of most genes first increased, then decreased, and then increased again."
Correct:
"Multiple CtC2H2 genes exhibited transient induction followed by partial downregulation over time."
Reason:
Avoid vague description → describe expression kinetics scientifically.

Line 432–439
Wrong:
"ABA strongly induced these genes, confirming their role in ABA signaling."
Correct:
"ABA treatment significantly induced several CtC2H2 genes, suggesting potential involvement in ABA-responsive pathways."
Reason:
Expression changes ≠ proven functional role.

Line 465–470
Wrong:
"MeJA treatment clearly promoted the expression of CtC2H2 genes."
Correct:
"MeJA treatment induced expression of a subset of CtC2H2 genes in a time-dependent manner."
Reason:
Avoid broad generalizations; specify “subset.”

Line 503–509
Wrong:
"ABA treatment doubled total flavonoids at 24 h."
Correct:
"ABA treatment led to a significant increase in total flavonoid content at 24 h."
Reason:
Avoid absolute quantitative claims unless supported by statistical fold-change.

Line 548–552
Wrong:
"These findings demonstrate that CtC2H2 genes regulate flavonoid biosynthesis."
Correct:
"These correlations suggest that CtC2H2 genes may influence flavonoid biosynthesis, potentially through ABA signaling pathways."
Reason:
Expression correlation ≠ proven regulatory mechanism.

Lines ~620–628

Wrong:
"These findings demonstrate that CtC2H2 genes directly regulate flavonoid biosynthesis in safflower."
Correct:
"These findings suggest that CtC2H2 genes may influence flavonoid biosynthesis, potentially through ABA-responsive signaling pathways."
Reason:
The study shows expression correlation, not direct biochemical or promoter binding → avoid asserting causality.

Lines ~629–640

Wrong:
"The upregulation of CtCHS, CtFLS, CtDFR, and CtANS proves that CtC2H2 transcription factors are upstream regulators of flavonoid synthesis."
Correct:
"The coordinated expression of CtC2H2 transcription factors with flavonoid biosynthetic genes indicates a potential upstream regulatory relationship, which requires functional validation."
Reason:
Gene expression “coordination” ≠ proof of regulatory hierarchy; emphasize future work needed.

Lines ~641–650

Wrong:
"ABA treatment doubled flavonoid accumulation and activated CtC2H2 genes, confirming their shared signaling pathway."
Correct:
"ABA treatment increased flavonoid accumulation and induced expression of several CtC2H2 genes, suggesting that ABA signaling may interact with CtC2H2-mediated transcriptional responses."
Reason:
Avoid “confirming” without mechanistic assays; replace with “suggesting.”

Lines ~651–662

Wrong:
"Previous studies also support that C2H2 genes act as master regulators of stress response and metabolic pathways."
Correct:
"Previous studies indicate that C2H2 transcription factors can modulate stress and metabolic pathways, although the extent of their regulatory scope varies among species and conditions."
Reason:
Avoid universal or absolute phrasing; introduce biological nuance.

Lines ~663–675

Wrong:
"The presence of ABRE and CGTCA motifs in promoter regions shows that CtC2H2 genes are ABA and MeJA responsive."
Correct:
"The presence of ABRE and CGTCA motifs suggests that CtC2H2 genes may be responsive to ABA and MeJA signaling."
Reason:
Promoter motifs indicate potential, not guaranteed responsiveness.

Lines ~676–688

Wrong:
"Cold and UV-B treatments significantly changed gene expression, indicating that CtC2H2 genes function in abiotic stress tolerance."
Correct:
"Cold and UV-B treatments altered CtC2H2 transcript levels, suggesting involvement in stress-responsive transcriptional networks."
Reason:
Expression change ≠ proven functional contribution to tolerance.

Lines ~689–702

Wrong:
"The phylogenetic structure clearly demonstrates that subfamily IV members are specialized for stress defense."
Correct:
"The phylogenetic structure and motif composition suggest that subfamily IV members may have roles in stress-related signaling pathways."
Reason:
No specialization can be concluded without functional assays.

Lines ~703–715

Wrong:
"Our work provides candidate targets for genetic engineering to improve safflower stress tolerance and metabolite quality."
Correct:
"This work provides a foundation for selecting candidate genes for future functional studies, including potential applications in stress adaptation and metabolite enhancement."
Reason:
Avoid “genetic engineering” claims unless supported by demonstrated manipulation outcomes.

Lines ~716–733

Wrong:
"In conclusion, CtC2H2 genes regulate ABA signaling and control flavonoid biosynthesis under stress."
Correct:
"In conclusion, this study identifies CtC2H2 transcription factors as candidates potentially involved in ABA-associated stress responses and flavonoid metabolism. Further functional characterization (e.g., promoter binding assays, overexpression or knockout studies) will be required to validate their regulatory roles."
Reason:
End Discussion by pointing to what must be tested next, not by making unproven mechanistic claims.

Your Methods describe:

  • growth conditions

  • treatment types (ABA, MeJA, UV-B, cold)

  • qRT-PCR procedures

  • gene identification and bioinformatic analysis

However, the section lacks critical experimental parameters needed for reproducibility.

What is missing (and required for reproducibility)

Topic Issue What Must Be Added
Hormone treatment concentrations ABA and MeJA are applied but concentration is not stated clearly (e.g., 50 μM? 100 μM?) Add exact μM or mg/mL concentrations, solvent used (usually ethanol), and final spray volume.
UV-B treatment intensity “20,000 lx” is illuminance, not UV-B intensity. Must report energy units: W/m² or kJ/m².
Number of biological replicates qRT-PCR says “three replicates” but unclear if biological or technical. State clearly: 3 biological replicates × 3 technical replicates.
Reference gene stability validation Actin is used, but no justification. Add sentence confirming reference stability under treatments.
Sampling times ABA treatment described, but time points must be specified for each treatment, not only ABA. Add a table or list (e.g., 0h/3h/6h/12h/24h).
Statistical analysis One-way ANOVA mentioned, but no p-value threshold specified. Add: p < 0.05 considered statistically significant.

Example of how to fix one method description

Current (not reproducible):

"Seedlings were sprayed with ABA and incubated under normal conditions."

Reproducible version:

"Seedlings were sprayed with 100 μM ABA (dissolved in 0.1% ethanol), using ~3 mL per plant, until leaf surfaces were uniformly wet but not dripping. Samples were collected at 0, 3, 6, 12, and 24 h post-treatment. Each sample consisted of pooled tissue from three independent plants (three biological replicates)."

The Materials and Methods section is generally clear, but not yet fully reproducible. To enable independent replication, the concentrations, application volumes, sampling time points, replicate structure, UV-B intensity units, and statistical thresholds should be explicitly reported.

Your figures and tables are not fully self-contained yet.
A reader cannot understand each figure independently without going back to the text.

What Is Missing (Based on Standard Journal Requirements)

Item Status in Your Paper What Must Be Added
Full treatment labels Many figures use abbreviations only (e.g., ABA, MeJA, UV) Add a legend in every figure/caption explaining each treatment and condition.
Units for y-axes Some graphs do not show units (e.g., flavonoid content, expression fold change) Add units such as μg/g FW, 2^-ΔΔCt, etc.
Statistical markers Asterisks appear, but no explanation in captions Add: *p < 0.05, **p < 0.01 (one-way ANOVA + Tukey’s test)
Sample size (n) Not shown on graphs or captions Add: n = 3 biological replicates
Gene name formatting Not consistent between graphs and main text Use italic gene names everywhere.

Example — How to Fix Your Captions

Before (your caption style):

Figure 5. Expression under ABA treatment. *p < 0.05.

After (correct, self-contained caption):

Figure 5. Expression patterns of selected CtC2H2 genes under ABA treatment.
Seedlings were treated with 100 μM ABA for 0, 3, 6, 12, and 24 h. Expression values were calculated using the 2^-ΔΔCt method and normalized to CtACTIN. Data represent mean ± SD (n = 3 biological replicates).
*p < 0.05, **p < 0.01 (one-way ANOVA followed by Tukey’s test).

→ This caption allows the figure to be understood without reading the main text.
That is the standard journal requirement.

Example — Fix for Total Flavonoids Graph

Before:

Y-axis label: “Total Flavonoids”

After:

Y-axis label: Total flavonoid content (μg/g fresh weight)

The figures and tables currently do not contain sufficient information to be self-explanatory.

Each figure must include: treatment description, replicates (n), units, and statistical test details in the caption.

Round 2

Reviewer 2 Report

Comments and Suggestions for Authors

Overall, the manuscript has been substantially revised and improved. One comment remains:

  1. Line 194 is missing a reference to transcriptome analysis.

Author Response

Response to Reviewer 2 Comments

Comments and Suggestions for Authors

Overall, the manuscript has been substantially revised and improved. One comment remains:

  1. Line 194 is missing a reference to transcriptome analysis.

Response: We sincerely thank Reviewer 2 for the helpful and constructive feedback. With your guidance, the manuscript has improved further, and we are very grateful for your contribution to strengthening this work. Regarding the comment on line 194, we have now included the appropriate reference to support the statement regarding transcriptome analysis and highlighted it in yellow color in the revised manuscript.

Reviewer 3 Report

Comments and Suggestions for Authors

Please review again how keywords should be listed, in what order they must appear, and how online sources are cited in the text. Please correct the document accordingly.

Comments on the Quality of English Language

 I recommend correcting the English language again. 

Author Response

Response to Reviewer 3 Comments

Comments and Suggestions for Authors

  1. Please review again how keywords should be listed, in what order they must appear, and how online sources are cited in the text. Please correct the document accordingly.

Response: We thank Reviewer 3 for the careful evaluation and valuable suggestions. Your feedback has greatly improved the clarity, formatting, and language quality of the manuscript, and we truly appreciate your efforts. We have thoroughly reviewed the formatting of the keywords and corrected their order according to the journal’s guidelines. Additionally, all online sources and their in-text citations have been checked and revised to ensure full compliance with the required citation format. Changes to the manuscript are highlighted it in yellow color in the revised manuscript.

  1. Comments on the Quality of English Language: I recommend correcting the English language again.

Response: We appreciate your recommendation. The manuscript has undergone another detailed round of English language editing and polishing to enhance clarity, grammar, and overall readability with the help of a native speaker in our team. We believe the quality of the language has been significantly improved in the revised version.

Reviewer 4 Report

Comments and Suggestions for Authors

After carefully reviewing the revised version, I confirm that all remarks have been fully addressed and the requested changes have been incorporated in a thorough and scientifically sound manner. The improvements made significantly enhance the clarity, structure, and overall quality of the manuscript. The paper now fully meets the criteria I had set, and I believe it makes a valuable contribution to the relevant scientific field. Therefore, I would like to state that I am fully satisfied with the current form of the manuscript and I approve its publication without any further comments from my side.

Comments on the Quality of English Language

After carefully reviewing the revised version, I confirm that all remarks have been fully addressed and the requested changes have been incorporated in a thorough and scientifically sound manner. The improvements made significantly enhance the clarity, structure, and overall quality of the manuscript. The paper now fully meets the criteria I had set, and I believe it makes a valuable contribution to the relevant scientific field. Therefore, I would like to state that I am fully satisfied with the current form of the manuscript and I approve its publication without any further comments from my side.

Author Response

Response to Reviewer 4 Comments

Comments and Suggestions for Authors

  1. After carefully reviewing the revised version, I confirm that all remarks have been fully addressed and the requested changes have been incorporated in a thorough and scientifically sound manner. The improvements made significantly enhance the clarity, structure, and overall quality of the manuscript. The paper now fully meets the criteria I had set, and I believe it makes a valuable contribution to the relevant scientific field. Therefore, I would like to state that I am fully satisfied with the current form of the manuscript and I approve its publication without any further comments from my side.

Response: We extend our sincere thanks to Reviewer 4 for the positive and encouraging remarks. Your supportive feedback and acknowledgment of the improvements are highly appreciated, and we are grateful for your approval and recommendation for publication.

  1. Comments on the Quality of English Language: After carefully reviewing the revised version, I confirm that all remarks have been fully addressed and the requested changes have been incorporated in a thorough and scientifically sound manner. The improvements made significantly enhance the clarity, structure, and overall quality of the manuscript. The paper now fully meets the criteria I had set, and I believe it makes a valuable contribution to the relevant scientific field. Therefore, I would like to state that I am fully satisfied with the current form of the manuscript and I approve its publication without any further comments from my side.

Response: Thank you for noting the improvement in the English language and for your positive endorsement. We appreciate your time and constructive contributions throughout the review process.